# Vietnamese health professionals' views on the status of the fetus and maternal and fetal health interests: A regional, cross-sectional study from the Hanoi area

Ingrid Mogren[1], Pham Thi Lan[2], Ho Dang Phuc[3], Sophia Holmlund[1,4,5], Rhonda Small[5], Joseph Ntaganira[6†], Jean Paul Semasaka Sengoma[1], Hussein Lesio Kidanto[7], Matilda Ngarina[8], Cecilia Bergström[1]*

1 Department of Clinical Sciences, Obstetrics and Gynecology, Umeå University, Umeå, Sweden, 2 Department of Dermatology and Venereology, Hanoi Medical University, Hanoi, Vietnam, 3 Institute of Mathematics, Vietnam Academy of Science and Technology, Hanoi, Vietnam, 4 Department of Nursing, Umeå University, Umeå, Sweden, 5 Judith Lumley Centre, School of Nursing and Midwifery, La Trobe University, Melbourne, Australia, 6 School of Public Health, College of Medicine and Health Sciences, University of Rwanda, Kigali, Rwanda, 7 Medical College, East Africa Aga Khan University, Dar es Salaam, Tanzania, 8 Department of Obstetrics and Gynaecology, Muhimbili National Hospital, Dar es Salaam, Tanzania

† Deceased.
* cecilia.bergstrom@umu.se

**Data Availability Statement:** Data cannot be shared publicly because of ethical and legal restrictions. Data are available from Hanoi Medical

## Abstract

Obstetric ultrasound is an important tool in managing pregnancies and its use is increasing globally. However, the status of the pregnant woman and the fetus may vary in terms of clinical management, views in the community and legislation. To investigate the views and experiences of Vietnamese health professionals on maternal and fetal health interests, priority setting and potential conflicts, we conducted a cross-sectional study using a structured questionnaire. Obstetricians/gynecologists, midwives and sonographers who manage pregnant women in maternity wards were invited to participate. We purposively chose public health facilities in the Hanoi region of Vietnam to obtain a representative sample. The final sample included 882 health professionals, of which 32.7% (n = 289) were obstetricians/gynecologists, 60.7% (n = 535) midwives and 6.6% (n = 58) sonographers. The majority of participants (60.3%) agreed that "The fetus is a person from the time of conception" and that maternal health interests should always be prioritised over fetal health interests in care provided (54.4%). 19.7% agreed that the fetus is never a patient, only the pregnant woman can be the patient, while 60.5% disagreed. Participants who performed ultrasounds were more likely to agree that fetal health interests are being given more weight in decision-making the further the gestation advances compared to those who did not perform ultrasounds (cOR 2.47, CI 1.27–4.79: n = 811). A significant proportion of health professionals in Vietnam assign the fetus the status of being a person, where personhood gradually evolves during pregnancy. While the fetus is often considered a patient with its own health interests, a majority of participants did give priority to maternal health interests. Health professionals appear to favour increased legal protection of the fetus. Strengthening the legal status of the

University, on behalf of the Hanoi Medical University Review Board (HMUIRB), for researchers who meet the criteria for access to confidential data. Please contact HMUIRB by e-mail: irb@hmu.edu.vn. Please note that data requests can be sent directly to the Hanoi Medical University Review Board (HMUIRB). Inquiries for data access can also be sent to Pham Thi Lin (lanphamthi009@gmail.com), who will then contact the ethics board for permission to share the data openly.

**Funding:** Financial support was received from Umeå University www.umu.se (IM), Västerbotten County Council www.regionvasterbotten.se (IM), the Swedish Research Council www.vr.se, Sweden (2014-2672) (IM), the Swedish Research Council for Health, Working Life and Welfare (FORTE) www.forte.se (IM). The funders had no role in study design, data collection and analysis, decision to publish, or preparation of the manuscript.

**Competing interests:** The authors have declared that no competing interests exist.

fetus might have adverse implications for maternal autonomy. Measures to restrict maternal autonomy might require close observation to ensure that maternal reproductive rights are protected.

## Background

Obstetric ultrasound is increasingly used in low- and middle-income countries [1]. The World Health Organization (WHO) recommends that all pregnant women access at least eight contacts with a healthcare provider during pregnancy, including one obstetric ultrasound before the 24[th] week of gestation [2]. Early obstetric ultrasound is performed to assess the location of the pregnancy, fetal cardiac activity, estimate gestational age, detect multiple pregnancy and fetal anomaly, reduce the induction of post-term pregnancy and improve the woman's pregnancy experience [3].

Guidelines issued by the Ministry of Health in Vietnam recommend at least four antenatal care (ANC) visits [4]. Although the average number of ANC visits varies between urban and rural areas, nearly all Vietnamese women attend ANC at least once during pregnancy [5]. In Vietnam, obstetric ultrasound has been an integral part of ANC since the early 21st-century [6, 7], where three routine obstetric ultrasound examinations are recommended with an additional third-trimester growth ultrasound [2]. Nevertheless, a considerable overuse of obstetric ultrasound among Vietnamese women in urban areas (averaging six ultrasounds) has been reported [8]. In contrast, women in rural areas have an average of 3.5 ultrasound examinations [4]. In Vietnam, congenital malformations are often seen as a burden to the family and society [6]. Thus, prenatal diagnosis of congenital anomalies in the second trimester has become a part of antenatal care [9].

The interpretation of personhood has been debated throughout history, with varying perspectives in philosophy, ethics and legislation. The philosopher John Locke proposed that the human is essentially a specific type of animal, which in turn is a being possessed of a living, organised body; on the other hand, a person is essentially an intelligent being characterised by reason and an understanding of itself as existing over time [10]. Peter Singer, influenced by Locke, argues that a being may be considered a person provided that four characteristics are exhibited. A rational and self-conscious being i) is aware of itself as an extended body existing over an extended period; ii) is desiring and plan-making; iii) contains as a necessary condition for the right to life that it wants to continue living; and iv) is autonomous being [10]. Applying the view of Singer, the fetus cannot be considered a person, and a newborn does not carry the full set of human rights as an older person. Health professionals may consider the fetus a person and patient, although conflicting perspectives on fetal status are evident [11]. It has been argued, however, that the designation of the fetus as a patient, i.e., as a medically treatable being, is not equivalent to attribution of personhood [11, 12] since personhood bears much more philosophical or theological weight–depending on one's beliefs–and requires considerably more than "treatability" to justify the attribution of personhood to the fetus [12]. Along with developments in fetal diagnostics and therapy, which have resulted in improvements in fetal health outcomes, the question of whether the fetus should be regarded as a person has been raised [13].

Studies on health professionals' experiences and views on different aspects of the fetus, the setting of priorities in relation to medical intervention and potential conflicts between maternal and fetal health interests are limited in the scientific literature. These areas are important

to study, both in relation to maternal autonomy/maternal health and potential conflicts between maternal autonomy/maternal health and fetal rights/health, since these may contribute to decisions made in clinical management. Therefore, we wanted to investigate if background factors such as current profession, age, gender, number of years in the current health profession, marital status, children of their own, type of healthcare provided (public/private/both), performing obstetric ultrasound examinations, and role in decision-making based on obstetric ultrasound influence health professionals' experiences and views on maternal and fetal health interests when providing maternity care in Vietnam. More specifically, we wanted to explore:

- Whether the fetus is considered a person before birth, and if so, at what point in time.

- Whether the fetus is considered a patient.

- How maternal and fetal health interests are viewed during the course of the pregnancy, and any potentially conflicting maternal and fetal health interests that may arise.

- Protection of the fetus by law.

This study is part of the large international CROCUS Study, aiming to investigate obstetricians' and midwives' experiences and views on the role of obstetric ultrasound in relation to clinical management, ethical aspects, and maternal and fetal interests in high-, middle- and low-income countries.

## Methods

### Study design, sampling and setting

The methods used have been described in detail in a previous publication from the Cross Country Ultrasound Study (CROCUS) [14]. In short, CROCUS is a two-phase project consisting of 1) a qualitative phase consisting of focus group discussion with midwives and in-depth interviews with obstetricians and 2) a quantitative phase utilising a questionnaire developed from the qualitative results in Phase 1 of the CROCUS [7, 15–23], and has been presented in detail in previous publications [14, 24]. This study is a cross-sectional study using the questionnaire for data collection purposes.

Eligible participants were obstetricians/gynecologists, midwives and sonographers. The public health facilities were purposively selected to obtain a representative sample of health professionals clinically managing pregnant women in the Hanoi region, Vietnam. One national hospital, one provincial hospital, 24 district hospitals and three maternity homes were included in the study, thus representing urban, semi-urban and rural areas and different levels of health facilities in the Hanoi region.

### Power calculation

Due to the limited literature, a power calculation was performed based on the estimated prevalence of outcome and background characteristics before data collection. The largest sample size required to detect a difference in the proportion of 0.10 with a power of 80% and a significance level of 5% was for the outcome variables "every woman should undergo ultrasound examination in pregnancy to determine gestational age" and the background variable "work experience over and under five years", a sample of 290 physicians and a corresponding number of midwives was required.

## Data collection tool–the questionnaire

The questionnaire consists of 105 questions and statements, all related to clinical management, ethical aspects, and maternal and fetal health. The research team developed the questionnaire in English, led by principal investigator Ingrid Mogren (IM), representing all six countries (Australia, Norway, Rwanda, Sweden, Tanzania, and Vietnam). The response options for each item were fixed, with some using Likert scales. The questionnaire used in Vietnam was translated into Vietnamese by a native Vietnamese speaker, not part of the research team. It was then back-translated by another external person into English, resulting only in small adjustments. The questionnaire was pilot-tested with ten obstetricians, six midwives and two sonographers, resulting in no further revisions. In this study, 18 statements were used as outcome variables (Table 1), and several selected background characteristics were used as predictor variables. The word fetus is used in the questionnaire as synonymous with the term embryo for pre-specified statements related to early pregnancy.

## Data collection procedures and participants

Prior to commencing data collection, ethical approval was obtained from the Hanoi Medical University Review Board in Bio-Medical Research (reference 141/HMU IRB). All methods were performed in accordance with the relevant guidelines and regulations. Participants were provided with verbal information about the study in Vietnamese. Participation was voluntary and no identifiable information was collected. Informed verbal consent was obtained from all subjects, witnessed by one of the four experienced data collectors. In addition, all participants received payment of 100,000 VND for their involvement in the study and signed a remuneration form that could not be traced to the questionnaire. The manuscript does not contain individual personal information; hence, obtaining consent for publication was deemed unnecessary. Data collection took place from April 10, 2017, to April 28, 2017 [14]. Eligible participants were health professionals caring for pregnant women in the maternity wards on

**Table 1. Outcome statements and response options[a] in the CROCUS questionnaire.**

*Statements on views of the embryo and the fetus*
  1. The fetus is a *person* from the time of conception
  2. The fetus is a *person* from the time heartbeats are detected
  3. The fetus is a *person* from the time the pregnant woman experiences fetal movements
  4. The fetus is a *person* when it can survive outside the uterus
  5. The fetus is a *person* when the pregnant woman considers it to be a person
  6. The fetus is a **not** a *person* until it is born
  7. Seeing the fetus through ultrasound makes me think of the fetus more as a person
  8. The fetus is a *patient* when the woman seeks health care for her pregnancy
  9. The fetus becomes a *patient* when fetal abnormalities are detected
  10. The fetus becomes a *patient* when the pregnant woman receives medical care to enhance fetal outcome(s)
  11. The fetus is never a *patient*, only the pregnant woman can be the patient

*Statements on maternal and fetal health interests in maternity care*
  12. Maternity care sometimes involves prioritising between maternal and fetal health interests
  13. The delivery sometimes has to be postponed in order to improve fetal outcome, although the pregnant woman may be at risk
  14. Maternal health interests should always be prioritised over fetal health interests in care provided
  15. Fetal health interests are being given more weight in decision-making, the further the gestation advances
  16. Fetal health interests are being given more consideration in care as opportunities for fetal diagnostic and treatment develop
  17. Fetal health interests are being given more consideration because of advances in neonatal care
  18. Fetal health interests should be better protected by law

[a]Response options: Strongly agree, Agee, Neutral, Disagree, Strongly disagree

the day of data collection at each study site. No eligible participant declined to participate in the study. The final sample included 882 health professionals. Two experienced data administrators at Hanoi Medical University entered data in an SPSS file.

## Independent variables

Health professions included obstetrician/gynaecologist, general practitioner, resident physician, physician other, midwife, radiologist/sonographer and 'other'. Health profession was after that categorised into three groups: obstetricians/gynaecologists, sonographers, and midwives. Resident physicians undergoing postgraduate training (n = 9) and general practitioners (n = 12) were also included in the category obstetricians/gynaecologists since they worked in the same departments and performed similar duties as the obstetricians/gynaecologists. One participant who was an anaesthesiologist by profession but was working with maternity care was also categorised as an obstetrician/gynaecologist. One nurse working in maternity care was categorised as a midwife. Age was calculated as a continuous variable using birth year and year of data collection. Years in the profession and years in health care were treated as continuous variables. Gender included male or female. Health facilities included a national hospital, a provincial hospital, district hospitals and maternity homes. Marital status included married, separated/divorced, widowed and unmarried/single. Having children included yes or no. Religious faith included the responses yes, no, and I prefer not to answer this question. Type of healthcare provided was classified as public, and both public and private healthcare. Area of health facility was categorised as hospitals in urban (n = 7), semi-urban (n = 5), and rural (n = 17) areas of Hanoi. Role in clinical decision-making and performing ultrasound examinations included the responses yes or no.

## Dependent variables

The dependent variables, i.e., the pre-specified statements with fixed response alternatives, are presented in Table 1. The term includes both the response options agree and strongly agree, and the term disagree includes both the response options disagree and strongly disagree.

## Statistics

Descriptive statistics were calculated. Tests of difference in mean values were assessed using Student's t-test, and tests of difference for categorical variables were assessed using Pearson's Chi-Square test with a p-value set at 0.05. Univariate and multivariate logistic regression analyses were performed for exposures and outcomes, calculating odds ratios (OR) and their 95% confidence intervals (CI) for possible associations. Logistic regression analysis for different statements excluded the response option "neutral" from the original five response categories, dividing the four remaining categories into two (agree/strongly agree and disagree/strongly disagree). In multivariate logistic regression, only variables that demonstrated significance in univariate logistic regression were included in the multivariate logistic regression models. Statements 1, 2 and 3 in the section "Statements on views of the fetus" (Table 1) were analysed in ascending chronological order, assuming that if the fetus were considered a person at an earlier chronologic stage, then personhood was presumed to be continuous during the remaining course of the pregnancy and at birth. The response categories agree and strongly agree were merged into one category. Venn diagrams illustrate similarities and differences in agreement/disagreement for selected statements. IBM SPSS vs. 28 was used in all analyses.

# Results

Table 2 presents the main background characteristics of the study sample. A total of 882 health professionals participated in the study, including 289 obstetricians/gynecologists (32.7%), 535 midwives (60.7%) and 58 sonographers (6.6%). The mean age of the participants was 34.8 years, the mean years in the profession was 10.3, and the majority of participants were female (81.4%). 32.8% reported performing ultrasound examinations, and 72.5% reported a role in clinical decision-making (Table 2). A small proportion of the participants reported having a religious faith (2.2%; n = 19), whereas the vast majority reported having no religious faith (95.9%; n = 836), and 1.9% of participants selected the response option "I prefer not to answer this question". Religion was not a discriminatory factor in any analysis. Sonographers were excluded from all logistic regression analyses as they represented only 6.6% of the total sample (n = 58). These results have been reported previously [14].

## Views of the fetus as a person

Fig 1 presents the distribution of separate and accumulated responses for statements 1, 2 and 3 (Table 1) regarding whether the fetus is considered a "person." 60.3% of participants agreed with statement 1, "The fetus is a person from the time of conception", 56.4% agreed with statement 2, "The fetus is a person from the time heartbeats are detected" and 42.1% agreed with statement 3, "The fetus is a person from the time the pregnant woman experiences fetal movements". 82.7% agreed with statements 1 and 2, and 87.2% agreed with statements 1, 2 and 3. 17.5% of participants agreed that "The fetus is not a person until it is born". A majority (54.6%) disagreed that "The fetus is a person when the pregnant woman considers it to be a person". Table 3 presents tests of difference with Pearson's Chi-Square test for background variables in relation to statements 1 and 6.

**The fetus is a person from the time of conception.** Significant differences in views about whether "the fetus is a person from the time of conception" were found for health professional category, gender, performing ultrasounds and having a role in clinical decision-making (Table 3). Participants performing ultrasound examinations were less likely to agree with the statement (crude odds ratio (cOR) 0.51; CI 0.35–0.72; n = 716)) than participants not performing ultrasounds. When adjusting for gender, the odds ratio remained unchanged; adjusted odds ratio (aOR) 0.51; CI 0.33–0.78; n = 716. In univariate logistic regression analysis, midwives were more likely to agree with the statement than obstetricians/gynecologists (cOR 2.37; CI 1.66–3.38; n = 673). In multivariate logistic regression, when adjusting for a) gender (aOR 2.32; CI 1.51–3.55; n = 673) and b) gender + performing ultrasounds (aOR 2.05; CI 1.07–3.92; n = 672), the odds ratio decreased slightly. Univariate and multivariate estimates and their 95% confidence intervals for health profession, gender and performing ultrasounds are comprehensively presented in Table 3.

**The fetus is not a person until it is born.** A minority (17.5%) agreed, whereas a majority (70.5%) disagreed that "the fetus is not a person until it is born". Significant background variables for this statement were working in both public and private health care, performing ultrasounds and having a role in decision-making (Table 3). Participants were more likely to disagree if they worked solely in public care (cOR 2.30; CI 1.11–4.76; n = 764) or did not perform ultrasounds (cOR 1.56; CI 1.08–2.27; n = 764).

**Seeing the fetus through ultrasound makes me think of the fetus more as a person.** Similar proportions agreed (40.4%;) and disagreed (41.1%;) with this statement. In univariate logistic regression analysis, participants performing ultrasounds were more likely to agree (cOR 1.46; CI 1.06–2.01) with the statement compared to health professionals not performing obstetric ultrasound examinations.

**Table 2. Background characteristics of the study sample (N = 882).**

| | All health professions | Obstetricians/Gynecologists | Sonographers[a] | Midwives |
|---|---|---|---|---|
| | N = 882 | n = 289 | n = 58 | n = 535 |
| **Age (years)** | **867 (98.3)** | **286 (99.0)** | **56 (96.6)** | **525 (98.1)** |
| Mean; SD[b] | 34.8; 8.7 | 36.6; 9.2 | 35.9; 8.8 | 33.7; 8.3 |
| Min-Max | 21–60 | 23–60 | 25–57 | 21–55 |
| **Years in profession** | **875 (99.2)** | **288 (99.7)** | **57 (98.3)** | **530 (99.1)** |
| Mean; SD[b] | 10.3; 8.3 | 10.4; 8.9 | 8.5; 7.2 | 10.5; 8.1 |
| Median | 8 | 7 | 7 | 9 |
| Min-max | 0–35 | 0–32 | 1–30 | 0.5–35 |
| **Years in health care** | **874 (99.1)** | **287 (99.3)** | **57 (98.3)** | **530 (99.1)** |
| Mean; SD[b] | 11.1; 8.5 | 11.6; 9.2 | 10.5; 8.1 | 10.9; 8.2 |
| Median | 9 | 9 | 10 | 9 |
| Min-max | 0–38 | 0–38 | 1–30 | 0.5–35 |
| **Gender** | **882 (100)** | **289 (100)** | **58 (100)** | **535 (100)** |
| Male | 164 (18.6) | 123 (42.6) | 41 (70.7) | - |
| Female | 718 (81.4) | 166 (57.4) | 17 (29.3) | 535 (100.0) |
| **Marital status** | **875 (99.2)** | **287 (99.3)** | **58 (100)** | **530 (99.1)** |
| Married | 759 (86.7) | 242 (84.3) | 45 (77.6) | 472 (89.1) |
| Separated/Divorced | 1 (0.1) | - | - | 1 (0.2) |
| Widowed | 4 (0.5) | 2 (0.7) | - | 2 (0.4) |
| Not married/Single | 111 (12.7) | 43 (15.0) | 13 (22.4) | 55 (10.4) |
| **Having children** | **879 (99.7)** | **288 (99.7)** | **58 (100)** | **533 (99.6)** |
| Yes | 727 (82.7) | 230 (79.9) | 43 (74.1) | 454 (85.2) |
| No | 152 (17.3) | 58 (20.1) | 15 (25.9) | 79 (14.8) |
| **Type of health care** | **881 (99.9)** | **289 (100)** | **58 (100)** | **534 (99.8)** |
| Public | 843 (95.7) | 268 (92.7) | 54 (93.1) | 521 (97.6) |
| Both public and private | 38 (4.3) | 21 (7.3) | 4 (6.9) | 13 (2.4) |
| **Level of health facility[c]** | **882 (100)** | **289 (100)** | **58 (100)** | **535 (100)** |
| National hospital | 152 (17.2) | 74 (25.6) | 8 (13.8) | 70 (13.1) |
| Provincial hospital | 194 (22.0) | 86 (29.8) | 10 (17.2) | 98 (18.3) |
| District hospital | 504 (57.1) | 121 (41.9) | 40 (69.0) | 343 (64.1) |
| Maternity home | 32 (3.6) | 8 (2.8) | - | 24 (4.5) |
| **Area of health facility[d]** | **882 (100)** | **289 (100)** | **58 (100)** | **535 (100)** |
| Urban | 470 (53.3) | 191 (66.1) | 31 (53.4) | 248 (46.4) |
| Semi-urban | 135 (15.3) | 35 (12.1) | 6 (10.3) | 94 (17.6) |
| Rural | 277 (31.4) | 63 (21.8) | 21 (36.2) | 193 (36.1) |
| **Role in clinical decision-making** | **840 (95.2)** | **281 (97.2)** | **57 (98.3)** | **502 (93.8)** |
| Yes, minor, moderate or major | 609 (72.5) | 259 (92.2) | 52 (91.2) | 298 (59.4) |
| No role in decision-making | 231 (27.5) | 22 (7.8) | 5 (8.8) | 204 (40.6) |
| **Performing ultrasound[e]** | **881 (99.9)** | **289 (100)** | **58 (100)** | **534 (99.8)** |
| Yes | 289 (32.8) | 228 (78.9) | 58 (100) | 3 (0.6) |
| No | 592 (67.2) | 61 (21.1) | - | 531 (99.4) |

Numbers in parenthesis are percentage unless otherwise specified.

[a]Sonographers are physicians

[b]SD = Standard Deviation

[c]Number of participants at specified health facilities

[d]Number of participants at specified areas of health facilities

[e]Performing obstetric ultrasound examinations

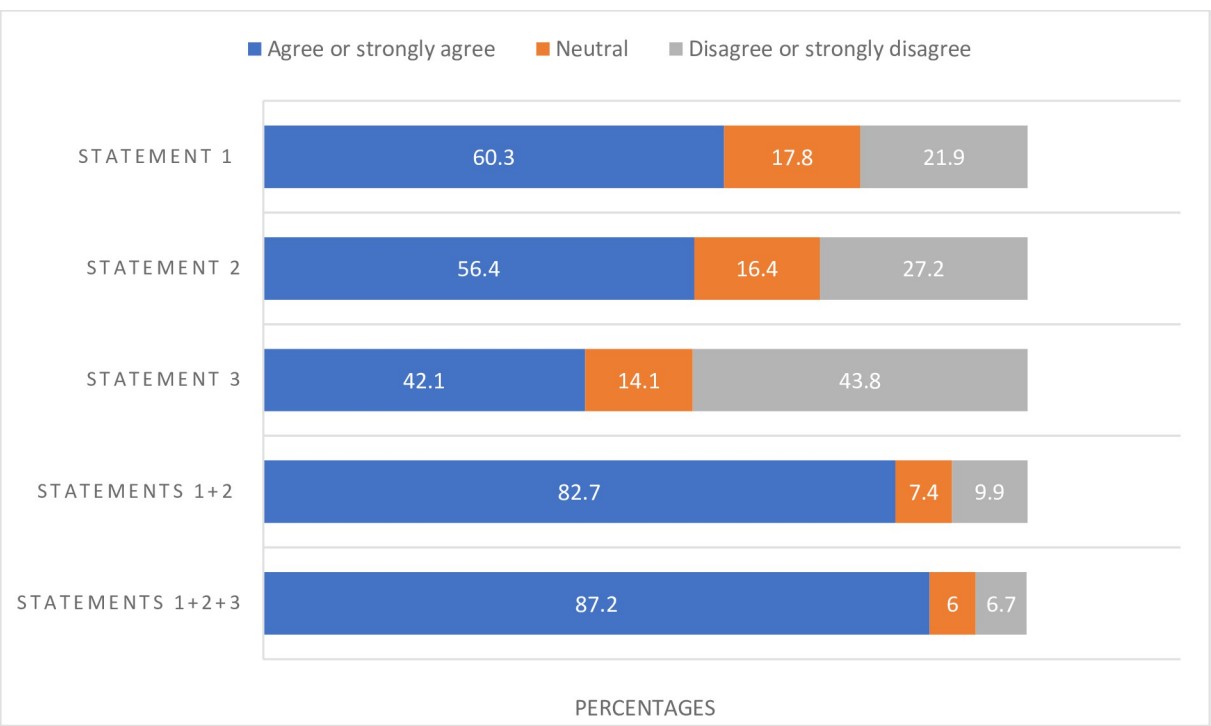

**Fig 1. Separated and accumulated agreements for statements 1–3.** Separated and accumulated agreements (%) for statement 1 ("The fetus is a person from the time of conception"; n = 872), statement 2 ("The fetus is a person from the time heartbeats are detected"; n = 877), and statement 3 ("The fetus is a person from the time the pregnant woman experiences fetal movements"; n = 877).

### Views of the fetus as a patient

**The fetus is a patient when the woman seeks health care for her pregnancy.** 27.6% agreed, and 57.9% disagreed that the fetus is a patient when the woman seeks health care for her pregnancy. Statistically significant background factors were health professional category, gender, working in public or public/private health care and performing ultrasound examinations (Table 4).

**The fetus becomes a patient when fetal abnormalities are detected.** In total, 45.5% agreed and 39.6% disagreed that the fetus becomes a patient when fetal abnormalities are detected. Statistically significant background factors were age, years in the profession, marital status, having children and having a role in clinical decision-making (Table 4; 3 response categories). In logistic regression analysis, participants reporting a major role in clinical decision-making demonstrated an increased likelihood of disagreeing compared to those who did not have a role in decision-making (cOR 1.48; CI 1.01–2.16).

**The fetus becomes a patient when the pregnant woman receives medical care to enhance fetal outcome(s).** Altogether, 39.8% agreed, and 41.4% disagreed that the fetus becomes a patient when the pregnant woman receives medical care to enhance fetal outcome(s). Background factors that were statistically significant were age, years in profession, marital status and having children (Table 4; 3 response categories).

**The fetus is never a patient, only the pregnant woman can be the patient.** 19.7% agreed and 60.5% disagreed with the statement. Statistically significant background factors were health professional category, age, gender, years in the profession, performing ultrasound examinations and having a role in clinical decision-making (Table 4). In logistic regression analysis, when only including obstetricians/gynecologists and midwives in the analysis, there

**Table 3. Health professionals' views on the embryo/fetus as a person in relation to background variables (N = 882).**

| | *The fetus is a person from the time of conception* | | | | *The fetus is not a person until it is born* | | | |
| | Agree or strongly agree | Disagree or strongly disagree | p-value[a] | | Agree or strongly agree | Disagree or strongly disagree | p-value[a] | |
| | | | 5 cat[b] | 3 cat.[c] | | | 5 cat[b] | 3 cat.[c] |
|---|---|---|---|---|---|---|---|---|
| **Health professionals** | | | | | | | | |
| Obstetricians/gynecologists | 135 (47.0) | 85 (29.6) | <0.001 | <0.001 | 58 (20.4) | 188 (66.0) | 0.172 | 0.299 |
| Midwives | 358 (67.9) | 95 (18.0) | | | 85 (16.1) | 386 (73.1) | | |
| Sonographers | 33 (56.9) | 11 (19.0) | | | 9 (16.1) | 39 (69.6) | | |
| **Age** | | | | | | | | |
| <35 years | 316 (60.0) | 113 (21.4) | 0.665 | 0.762 | 83 (15.7) | 377 (71.1) | 0.105 | 0.119 |
| ≥35 years | 201 (60.9) | 74 (22.4) | | | 67 (20.7) | 223 (68.8) | | |
| **Gender** | | | | | | | | |
| Male | 82 (50.6) | 44 (27.2) | 0.002 | 0.02 | 32 (20.1) | 101 (63.5) | 0.237 | 0.072 |
| Female | 444 (62.5) | 147 (20.7) | | | 120 (16.9) | 512 (72.1) | | |
| **Years in profession** | | | | | | | | |
| ≤10 years | 331 (58.2) | 130 (22.8) | 0.301 | 0.262 | 95 (16.7) | 403 (70.7) | 0.426 | 0.614 |
| >10 years | 189 (63.9) | 60 (20.3) | | | 55 (18.8) | 205 (70.2) | | |
| **Marital status[d]** | | | | | | | | |
| Married | 458 (61.1) | 159 (21.2) | 0.065 | 0.566 | 135 (18.1) | 526 (70.4) | 0.42 | 0.175 |
| Not married/single | 62 (55.9) | 27 (24.3) | | | 13 (11.8) | 80 (72.7) | | |
| **Having children** | | | | | | | | |
| Yes | 438 (61.0) | 155 (21.6) | 0.315 | 0.557 | 135 (18.9) | 498 (69.6) | 0.175 | 0.051 |
| No | 85 (56.3) | 36 (23.8) | | | 16 (10.7) | 113 (75.3) | | |
| **Public/Private health care[e]** | | | | | | | | |
| Public | 504 (60.4) | 183 (21.9) | 0.866 | 0.819 | 140 (16.8) | 590 (70.8) | 0.045 | 0.013 |
| Both public and private | 21 (56.8) | 8 (21.6) | | | 12 (34.3) | 22 (62.9) | | |
| **Performing ultrasound[f]** | | | | | | | | |
| Yes | 140 (48.8) | 80 (27.9) | <0.001 | <0.001 | 60 (21.2) | 180 (63.6) | 0.009 | 0.008 |
| No | 385 (65.9) | 111 (19.0) | | | 92 (15.7) | 432 (73.8) | | |
| **Role in clinical decision-making** | | | | | | | | |
| Yes | 355 (58.8) | 142 (23.5) | 0.716 | 0.385 | 102 (17.0) | 423 (70.5) | 0.039 | 0.728 |
| No | 146 (63.8) | 45 (19.7) | | | 41 (18.1) | 162 (71.4) | | |
| **Role in clinical decision-making, grade[g]** | | | | | | | | |
| Yes, a minor to moderate role | 246 (57.6) | 101 (23.7) | 0.002 | 0.548 | 71 (16.8) | 301 (71.3) | 0.299 | 0.724 |
| Yes, a major role | 109 (61.6) | 41 (23.2) | | | 31 (17.4) | 122 (68.5) | | |

Numbers in parenthesis are percentage unless otherwise specified.

[a]Pearson's Chi-Square test for comparison of difference between categories.

[b]The five categories of responses were included as separate categories in analysis: "Strongly agree", "Agree", "Neutral", "Disagree", "Strongly disagree". Response option "neutral" not presented in the table.

[c]The five categories of responses were categorised into three groups in analysis: "Strongly agree/agree", "Neutral", "Strongly disagree/Disagree" in analyses. Response option "neutral" not presented in the table.

[d]Marital status, dichotomous variable. The category not married/single do not include participants who reported that they were separated/divorced or widowed.

[e]Included health facilities were primarily offering public health care.

[f]Performing ultrasound examinations.

[g]Included in analysis were only participants reporting a role in clinical decision-making.

**Table 4. Health professionals' views on the fetus as a patient in relation to background variables (N = 882).**

| | The fetus is a patient when the woman seeks health care for her pregnancy | | | | The fetus becomes a patient when fetal abnormalities are detected | | | |
|---|---|---|---|---|---|---|---|---|
| | Agree or strongly agree | Disagree or strongly disagree | p-value[a] | | Agree or strongly agree | Disagree or strongly disagree | p-value[a] | |
| | | | 5 cat[b] | 3 cat[c] | | | 5 cat[b] | 3 cat[c] |
| **Health professional** | | | | | | | | |
| Obstetricians/gynecologists | 67 (23.4) | 176 (61.5) | 0.001 | 0.107 | 132 (46.0) | 111 (38.7) | 0.349 | 0.939 |
| Midwives | 155 (29.4) | 301 (57.1) | | | 237 (44.9) | 214 (40.5) | | |
| Sonographers | 18 (31.0) | 27 (46.6) | | | 28 (49.1) | 20 (35.1) | | |
| **Age** | | | | | | | | |
| <35 years | 143 (27.0) | 304 (57.4) | 0.629 | 0.429 | 240 (45.5) | 198 (37.5) | 0.057 | 0.029 |
| ≥35 years | 95 (29.1) | 190 (58.3) | | | 154 (46.8) | 140 (42.6) | | |
| **Gender** | | | | | | | | |
| Male | 45 (27.8) | 93 (57.4) | 0.001 | 0.991 | 81 (50.3) | 57 (35.4) | 0.464 | 0.385 |
| Female | 195 (27.5) | 411 (58.0) | | | 316 (44.4) | 288 (40.5) | | |
| **Years in profession** | | | | | | | | |
| ≤10 years | 148 (25.9) | 330 (57.7) | 0.086 | 0.064 | 246 (43.2) | 223 (39.1) | 0.01 | 0.003 |
| >10 years | 89 (30.5) | 171 (58.6) | | | 148 (50.2) | 120 (40.7) | | |
| **Marital status[d]** | | | | | | | | |
| Married | 206 (27.5) | 430 (57.5) | 0.346 | 0.631 | 349 (46.5) | 301 (40.1) | 0.018 | 0.004 |
| Not married/single | 28 (25.2) | 69 (62.2) | | | 44 (40.0) | 38 (34.5) | | |
| **Having children** | | | | | | | | |
| Yes | 198 (27.7) | 408 (57.0) | 0.24 | 0.286 | 330 (45.9) | 294 (40.9) | 0.013 | 0.011 |
| No | 39 (25.7) | 96 (63.2) | | | 66 (43.7) | 51 (33.8) | | |
| **Public/Private health care[e]** | | | | | | | | |
| Public | 231 (27.7) | 485 (58.2) | 0.012 | 0.195 | 381 (45.7) | 326 (39.1) | 0.541 | 0.355 |
| Both public and private | 9 (25.0) | 18 (50.0) | | | 16 (43.2) | 18 (48.6) | | |
| **Performing ultrasound[f]** | | | | | | | | |
| Yes | 68 (23.8) | 167 (58.4) | 0.001 | 0.062 | 133 (46.5) | 110 (38.5) | 0.562 | 0.884 |
| No | 172 (29.5) | 337 (57.7) | | | 263 (45.0) | 235 (40.2) | | |
| **Role in clinical decision-making** | | | | | | | | |
| Yes | 174 (29.0) | 344 (57.2) | 0.373 | 0.476 | 282 (46.8) | 232 (38.5) | 0.104 | 0.53 |
| No | 59 (25.8) | 132 (57.6) | | | 98 (43.2) | 97 (42.7) | | |
| **Role in clinical decision-making, grade[g]** | | | | | | | | |
| Yes, a minor to moderate role | 115 (27.1) | 251 (59.2) | 0.096 | 0.266 | 209 (49.1) | 153 (35.9) | 0.034 | 0.124 |
| Yes, a major role | 59 (33.3) | 93 (52.5) | | | 73 (41.2) | 79 (44.6) | | |
| | The fetus becomes a patient when the pregnant woman receives medical care to enhance fetal outcome(s) | | | | The fetus is never a patient, only the pregnant woman can be the patient | | | |
| | Agree or strongly agree | Disagree or strongly disagree | p-value[a] | | Agree or strongly agree | Disagree or strongly disagree | p-value[a] | |
| | | | 5 cat[b] | 3 cat[c] | | | 5 cat[b] | 3 cat[c] |
| **Health professional** | | | | | | | | |
| Obstetricians/gynecologists | 115 (40.5) | 115 (40.5) | 0.537 | 0.902 | 41 (14.3) | 181 (63.3) | 0.004 | 0.001 |
| Midwives | 205 (38.8) | 222 (42.5) | | | 127 (24.1) | 301 (57.0) | | |
| Sonographers | 26 (44.8) | 23 (39.7) | | | 3 (5.4) | 44 (78.6) | | |
| **Age** | | | | | | | | |
| <35 years | 201 (37.9) | 206 (38.9) | 0.001 | 0.001 | 109 (20.6) | 302 (57.2) | 0.099 | 0.027 |
| ≥35 years | 139 (42.8) | 148 (45.5) | | | 55 (16.8) | 217 (66.4) | | |

*(Continued)*

**Table 4.** (Continued)

| | | | | | | | | |
|---|---|---|---|---|---|---|---|---|
| **Gender** | | | | | | | | |
| Male | 68 (42.0) | 66 (40.7) | 0.159 | 0.77 | 19 (11.9) | 108 (67.5) | 0.064 | 0.021 |
| Female | 278 (39.3) | 294 (41.5) | | | 152 (21.4) | 418 (58.9) | | |
| **Years in profession** | | | | | | | | |
| ≤10 years | 211 (36.8) | 232 (40.5) | 0.001 | 0.001 | 119 (20.9) | 323 (56.7) | 0.024 | 0.008 |
| >10 years | 133 (45.9) | 124 (42.8) | | | 51 (17.4) | 197 (67.2) | | |
| **Marital status**[d] | | | | | | | | |
| Married | 298 (39.9) | 318 (42.6) | 0.041 | 0.023 | 152 (20.3) | 454 (60.6) | 0.229 | 0.163 |
| Not married/single | 43 (38.7) | 37 (33.3) | | | 16 (14.7) | 65 (59.6) | | |
| **Having children** | | | | | | | | |
| Yes | 283 (39.6) | 309 (43.2) | 0.04 | 0.015 | 146 (20.4) | 433 (60.4) | 0.526 | 0.387 |
| No | 61 (40.1) | 51 (33.6) | | | 25 (16.7) | 90 (60.0) | | |
| **Public/Private health care**[e] | | | | | | | | |
| Public | 330 (39.6) | 347 (41.6) | 0.883 | 0.792 | 167 (20.0) | 499 (59.8) | 0.488 | 0.223 |
| Both public and private | 16 (44.4) | 13 (36.1) | | | 4 (11.4) | 26 (74.3) | | |
| **Performing ultrasound**[f] | | | | | | | | |
| Yes | 120 (42.1) | 115 (40.4) | 0.426 | 0.568 | 35 (12.3) | 190 (66.9) | 0.005 | 0.001 |
| No | 225 (38.5) | 245 (42.0) | | | 136 (23.2) | 335 (57.3) | | |
| **Role in clinical decision-making** | | | | | | | | |
| Yes | 252 (42.1) | 240 (40.1) | 0.175 | 0.127 | 115 (19.1) | 361 (60.1) | 0.212 | 0.594 |
| No | 79 (34.5) | 101 (44.4) | | | 45 (19.8) | 142 (62.6) | | |
| **Role in clinical decision-making, grade**[g] | | | | | | | | |
| Yes, a minor to moderate role | 184 (43.7) | 159 (37.8) | 0.041 | 0.21 | 88 (20.9) | 238 (56.4) | 0.007 | 0.019 |
| Yes, a major role | 68 (38.2) | 81 (45.4) | | | 27 (15.1) | 123 (68.7) | | |

Numbers in parenthesis are percentage unless otherwise specified.

[a]Pearson's Chi-Square test for comparison of difference between categories.

[b]The five categories of responses were included as separate categories in analysis: "Strongly agree", "Agree", "Neutral", "Disagree", "Strongly disagree". Response option "neutral" not presented in the table.

[c]The five categories of responses were categorised into three groups in analysis: "Strongly agree/agree", "Neutral", "Strongly disagree/Disagree" in analyses. Response option "neutral" not presented in the table.

[d]Marital status, dichotomous variable. The category not married/single do not include participants who reported that they were separated/divorced or widowed.

[e]Included health facilities were primarily offering public health care.

[f]Performing ultrasound examinations.

[g]Included in analysis were only participants reporting a role in clinical decision-making.

was an increased likelihood of obstetricians/gynecologists disagreeing compared with midwives (cOR 1.86; CI 1.25–2.78; n = 650). Male participants were more likely to disagree than female participants (cOR 2.07; CI 1.22–3.49; n = 697). When adjusting for health profession, the association became non-significant (aOR 0.97; CI 0.48–1.92; n = 650). When adjusting health profession for gender, the odds ratio increased slightly (aOR 1.89; CI 1.14–3.11; n = 650). In logistic regression analysis, years in the profession were close to being statistically significant, where more than ten years in the profession indicated an increased probability of disagreeing with the statement (cOR 1.42; CI 0.98–2.07; n = 690; p = 0.064). Participants performing ultrasound examinations were more likely to disagree (cOR 2.20; CI 1.46–3.33; n = 696), and when adjusting for gender, the odds ratios remained significant (aOR 1.96; CI 1.19–3.21; n = 696).

### Maternal and fetal health interests in maternity care

**Maternity care sometimes involves prioritising between maternal and fetal health interests.** The vast majority (90.2%) agreed, with 5.2% disagreeing that maternity care sometimes involves prioritising between maternal and fetal health interests. Having a role in clinical decision-making was the only discriminating background factor (5 response categories, p-value = 0.021; 3 response categories, p-value = 0.099; Table 5); however, this was not significant in logistic regression analysis.

**The delivery sometimes has to be postponed in order to improve fetal outcome, although the pregnant woman may be at risk.** Similar proportions (40.0% and 39.8%) agreed and disagreed, respectively, that the delivery sometimes has to be postponed to improve fetal outcome, although the pregnant woman may be at risk. Significant background factors were health professional category, age and years in the profession (Table 5). When only including obstetricians/gynecologists and midwives in the logistic regression analysis, midwives were more likely to agree with the statement than obstetricians/gynecologists (cOR 1.52; CI 1.10–2.09; n = 654). When adjusting for gender, the adjusted odds ratio increased further (aOR 2.02; CI 1.35–3.00; n = 654).

**Maternal health interests should always be prioritised over fetal health interests in care provided.** A majority of participants (54.4%) agreed that maternal health interests should always be prioritised over fetal health interests in care provided, whereas 18.7% disagreed and 27.1% were neutral. There was no significant difference when comparing the views of obstetricians/gynecologists and midwives (p = 0.063). No background factor was found to be significant (Pearson's Chi-Square test for 5 and 3 response categories; data not presented in the table).

**Fetal health interests are being given more weight in decision-making, the further the gestation advances.** Most participants (88.2%) agreed that fetal health interests are being given more weight in decision-making the further the gestation advances, whereas a small proportion (4.2%) disagreed and 7.6% were neutral. Significant background factors were health profession category, age, gender and whether performing ultrasound examinations (Table 6). Midwives were more likely to agree with the statement than physicians (cOR 2.58, CI 1.27–5.21; n = 763). Participants who performed ultrasounds were also more likely to agree than those who did not (cOR 2.47, CI 1.27–4.79: n = 811).

**Fetal health interests are being given more consideration in care as opportunities for fetal diagnostics and treatment develop.** Nearly all participants (96.9%) agreed that fetal health interests are being given more consideration in care as fetal diagnostics and treatment opportunities develop, whereas very few (0.8%) disagreed and 2.3% were neutral. Although few participants disagreed, a significant background factor was having a role in clinical decision-making, where the likelihood of agreeing was increased for participants reporting that they had a role in clinical decision-making (Table 6; 3 response categories).

**Fetal health interests are being given more consideration because of advances in neonatal care.** The vast majority of participants (95.0%) agreed that fetal health interests are being given more consideration because of advances in neonatal care, whereas few (1.7%) disagreed and 3.3% were neutral.

**Fetal health interests should be better protected by law.** Most participants (89.0%) agreed that fetal health interests should be better protected by law, with only a few (1.1%) disagreeing and 9.8% were neutral. Although few participants disagreed with the statement, health professional category, workplace, and whether performing ultrasound examinations and having a role in clinical decision-making were associated with agreement/disagreement with the statement (Table 6).

**Table 5. Health professionals' views on maternal and fetal health interests in maternity care in relation to background variables (N = 882).**

| | Maternity care sometimes involves prioritising between maternal and fetal health interests | | | | The delivery sometimes has to be postponed in order to improve fetal outcome, although the pregnant woman may be at risk | | | |
| --- | --- | --- | --- | --- | --- | --- | --- | --- |
| | Agree or strongly agree | Disagree or strongly disagree | p-value[a] | | Agree or strongly agree | Disagree or strongly disagree | p-value[a] | |
| | | | 5 cat[b] | 3 cat[c] | | | 5 cat[b] | 3 cat[c] |
| **Health professional** | | | | | | | | |
| Obstetricians/gynecologists | 264 (92.0) | 13 (4.5) | 0.76 | 0.614 | 104 (36.6) | 141 (49.6) | 0.001 | 0.001 |
| Midwives | 477 (89.3) | 31 (5.8) | | | 28 (49.1) | 12 (21.1) | | |
| Sonographers | 51 (89.5) | 2 (3.5) | | | 216 (40.9) | 193 (36.6) | | |
| **Age** | | | | | | | | |
| <35 years | 482 (90.3) | 27 (5.1) | 0.372 | 0.903 | 213 (40.5) | 193 (36.7) | 0.004 | 0.022 |
| ≥35 years | 296 (89.7) | 19 (5.8) | | | 129 (39.3) | 146 (44.5) | | |
| **Gender** | | | | | | | | |
| Male | 148 (91.4) | 6 (3.7) | 0.448 | 0.61 | 76 (47.2) | 58 (36.0) | 0.305 | 0.113 |
| Female | 644 (89.9) | 40 (5.6) | | | 272 (38.4) | 288 (40.7) | | |
| **Years in profession** | | | | | | | | |
| ≤10 years | 512 (89.0) | 31 (5.4) | 0.171 | 0.154 | 233 (41.1) | 208 (36.7) | 0.006 | 0.016 |
| >10 years | 273 (92.2) | 15 (5.1) | | | 113 (38.3) | 135 (45.8) | | |
| **Marital status[d]** | | | | | | | | |
| Married | 679 (89.9) | 42 (5.6) | 0.371 | 0.42 | 307 (41.1) | 296 (39.6) | 0.134 | 0.241 |
| Not married/single | 102 (91.9) | 3 (2.7) | | | 38 (34.5) | 44 (40.0) | | |
| **Having children** | | | | | | | | |
| Yes | 651 (90.0) | 40 (5.5) | 0.3 | 0.67 | 295 (41.3) | 284 (39.7) | 0.141 | 0.116 |
| No | 138 (90.8) | 6 (3.9) | | | 52 (34.4) | 60 (39.7) | | |
| **Public/Private health care[e]** | | | | | | | | |
| Public | 757 (90.2) | 43 (5.1) | 0.505 | 0.651 | 332 (40.0) | 333 (40.1) | 0.484 | 0.759 |
| Both public and private | 34 (89.5) | 3 (7.9) | | | 15 (40.5) | 13 (35.1) | | |
| **Performing ultrasound[f]** | | | | | | | | |
| Yes | 529 (89.5) | 33 (5.6) | 0.851 | 0.617 | 242 (41.4) | 220 (37.6) | 0.301 | 0.177 |
| No | 262 (91.6) | 13 (4.5) | | | 106 (37.5) | 125 (44.2) | | |
| **Role in clinical decision-making** | | | | | | | | |
| Yes | 551 (91.1) | 25 (4.1) | 0.021 | 0.099 | 229 (38.1) | 250 (41.6) | 0.234 | 0.375 |
| No | 203 (87.9) | 18 (7.8) | | | 99 (43.2) | 85 (37.1) | | |
| **Role in clinical decision-making, grade[g]** | | | | | | | | |
| Yes, a minor to moderate role | 391 (91.6) | 17 (4.0) | 0.125 | 0.788 | 158 (37.4) | 170 (40.2) | 0.051 | 0.124 |
| Yes, a major role | 160 (89.9) | 8 (4.5) | | | 71 (39.9) | 80 (44.9) | | |

Numbers in parenthesis are percentage unless otherwise specified.

[a]Pearson's Chi-Square test for comparison of difference between categories.

[b]The five categories of responses were included as separate categories in analysis: "Strongly agree", "Agree", "Neutral", "Disagree", "Strongly disagree". Response option "neutral" not presented in the table.

[c]The five categories of responses were categorised into three groups in analysis: "Strongly agree/agree", "Neutral", "Strongly disagree/Disagree" in analyses. Response option "neutral" not presented in the table.

[d]Marital status, dichotomous variable. The category not married/single do not include participants who reported that they were separated/divorced or widowed.

[e]Included health facilities were primarily offering public health care.

[f]Performing ultrasound examinations.

[g]Included in analysis were only participants reporting a role in clinical decision-making.

**Table 6. Health professionals' views on maternal and fetal health interests in maternity care in relation to background variables (N = 882).**

| | Fetal health interests are being given more weight in decision-making, the further the gestation advances | | | | Fetal health interests are being given more consideration in care as opportunities for fetal diagnostic and treatment develop | | | |
|---|---|---|---|---|---|---|---|---|
| | Agree or strongly agree | Disagree or strongly disagree | p-value[a] 5 cat[b] | 3 cat[c] | Agree or strongly agree | Disagree or strongly disagree | p-value[a] 5 cat[b] | 3 cat[c] |
| **Health professional** | | | | | | | | |
| Obstetricians/gynecologists | 232 (80.0) | 18 (6.3) | <0.001 | <0.001 | 278 (97.2) | 2 (0.7) | 0.816 | 0.936 |
| Midwives | 498 (93.3) | 15 (2.8) | | | 517 (96.6) | 5 (0.9) | | |
| Sonographers | 45 (77.6) | 4 (6.9) | | | 57 (98.3) | - | | |
| **Age** | | | | | | | | |
| <35 years | 482 (90.4) | 18 (3.4) | 0.017 | 0.025 | 517 (97.0) | 6 (1.1) | 0.446 | 0.238 |
| ≥35 years | 279 (84.3) | 18 (5.4) | | | 320 (96.7) | 1 (0.3) | | |
| **Gender** | | | | | | | | |
| Male | 127 (78.4) | 3 (8.0) | <0.001 | <0.001 | 157 (97.5) | 1 (0.6) | 0.461 | 0.891 |
| Female | 648 (90.4) | 24 (3.3) | | | 695 (96.8) | 6 (0.8) | | |
| **Years in profession** | | | | | | | | |
| ≤10 years | 512 (89.2) | 21 (3.7) | 0.202 | 0.327 | 554 (96.5) | 6 (1.0) | 0.604 | 0.494 |
| >10 years | 256 (85.9) | 16 (5.4) | | | 291 (97.7) | 1 (0.3) | | |
| **Marital status[d]** | | | | | | | | |
| Married | 666 (88.0) | 31 (4.1) | 0.912 | 0.834 | 734 (97.1) | 5 (0.7) | 0.772 | 0.434 |
| Not married/single | 98 (89.1) | 5 (4.5) | | | 106 (95.5) | 2 (1.8) | | |
| **Having children** | | | | | | | | |
| Yes | 637 (87.9) | 30 (4.1) | 0.422 | 0.675 | 703 (97.1) | 4 (0.6) | 0.333 | 0.195 |
| No | 135 (89.4) | 7 (4.6) | | | 146 (96.1) | 3 (2.0) | | |
| **Public/Private health care[e]** | | | | | | | | |
| Public | 741 (88.2) | 35 (4.2) | 0.327 | 0.944 | 813 (96.8) | 7 (0.8) | 0.776 | 0.533 |
| Both public and private | 33 (86.8) | 2 (5.3) | | | 38 (100) | - | | |
| **Performing ultrasound[f]** | | | | | | | | |
| Yes | 232 (80.8) | 19 (6.6) | <0.001 | <0.001 | 280 (97.6) | 2 (0.7) | 0.495 | 0.738 |
| No | 542 (91.7) | 18 (3.0) | | | 571 (96.6) | 5 (0.8) | | |
| **Role in clinical decision-making** | | | | | | | | |
| Yes | 534 (88.0) | 23 (3.8) | 0.052 | 0.674 | 595 (98.0) | 3 (0.5) | 0.063 | 0.02 |
| No | 205 (89.1) | 10 (4.3) | | | 217 (94.3) | 3 (1.3) | | |
| **Role in clinical decision-making, grade[g]** | | | | | | | | |
| Yes, a minor to moderate role | 378 (88.3) | 14 (3.3) | 0.755 | 0.577 | 417 (97.4) | 3 (0.7) | 0.108 | 0.25 |
| Yes, a major role | 156 (87.2) | 9 (5.0) | | | 178 (99.4) | - | | |
| | Fetal health interests are being given more consideration because of advances in neonatal | | | | Fetal health interests should be better protected by law | | | |
| | Agree or strongly agree | Disagree or strongly disagree | p-value[a] 5 cat[b] | 3 cat[c] | Agree or strongly agree | Disagree or strongly disagree | p-value[a] 5 cat[b] | 3 cat[c] |
| **Health professional** | | | | | | | | |
| Obstetricians/gynecologists | 274 (95.8) | 4 (1.4) | 0.067 | 0.216 | 238 (83.5) | 2 (0.7) | 0.001 | 0.001 |
| Midwives | 508 (95.1) | 10 (1.9) | | | 50 (87.7) | 1 (1.8) | | |
| Sonographers | 52 (89.7) | 1 (1.7) | | | 490 (92.1) | 7 (1.3) | | |
| **Age** | | | | | | | | |
| <35 years | 505 (94.7) | 10 (1.9) | 0.734 | 0.924 | 476 (89.5) | 6 (1.1) | 0.703 | 0.823 |
| ≥35 years | 314 (95.2) | 5 (1.5) | | | 289 (88.1) | 4 (1.2) | | |

*(Continued)*

**Table 6.** (Continued)

| | | | | | | | | |
|---|---|---|---|---|---|---|---|---|
| **Gender** | | | | | | | | |
| Male | 151 (93.2) | 3 (1.9) | 0.021 | 0.428 | 136 (84.0) | 2 (1.2) | 0.115 | 0.061 |
| Female | 683 (95.4) | 12 (1.7) | | | 642 (90.2) | 8 (1.1) | | |
| **Years in profession** | | | | | | | | |
| ≤10 years | 540 (93.9) | 10 (1.7) | 0.065 | 0.065 | 504 (88.1) | 7 (1.2) | 0.807 | 0.566 |
| >10 years | 287 (97.0) | 5 (1.7) | | | 267 (90.5) | 3 (1.0) | | |
| **Marital status[d]** | | | | | | | | |
| Married | 715 (94.7) | 14 (1.9) | 0.662 | 0.501 | 674 (89.5) | 9 (1.2) | 0.481 | 0.337 |
| Not married/single | 108 (97.3) | 1 (0.9) | | | 93 (85.3) | 1 (0.9) | | |
| **Having children** | | | | | | | | |
| Yes | 143 (94.1) | 4 (2.6) | 0.487 | 0.632 | 649 (89.9) | 7 (1.0) | 0.163 | 0.145 |
| No | 688 (95.2) | 11 (1.5) | | | 126 (84.6) | 3 (2.0) | | |
| **Public/Private health care[e]** | | | | | | | | |
| Public | 799 (95.2) | 14 (1.7) | 0.238 | 0.24 | 750 (89.7) | 9 (1.1) | 0.013 | 0.006 |
| Both public and private | 34 (89.5) | 1 (2.6) | | | 27 (73.0) | 1 (2.7) | | |
| **Performing ultrasound[f]** | | | | | | | | |
| Yes | 270 (94.4) | 5 (1.7) | 0.215 | 0.822 | 241 (84.9) | 3 (1.1) | 0.004 | 0.011 |
| No | 563 (95.3) | 10 (1.7) | | | 537 (91.2) | 7 (1.2) | | |
| **Role in clinical decision-making** | | | | | | | | |
| Yes | 577 (95.4) | 10 (1.7) | 0.232 | 0.448 | 528 (87.7) | 3 (0.5) | 0.002 | 0.001 |
| No | 216 (93.5) | 4 (1.7) | | | 212 (92.2) | 6 (2.6) | | |
| **Role in clinical decision-making, grade[g]** | | | | | | | | |
| Yes, a minor to moderate role | 409 (95.8) | 7 (1.6) | 0.441 | 0.669 | 374 (88.4) | 3 (0.7) | 0.658 | 0.306 |
| Yes, a major role | 168 (94.4) | 3 (1.7) | | | 154 (86.0) | - | | |

Numbers in parenthesis are percentage unless otherwise specified.

[a]Pearson's Chi-Square test for comparison of difference between categories.

[b]The five categories of responses were included as separate categories in analysis: "Strongly agree", "Agree", "Neutral", "Disagree", "Strongly disagree". Response option "neutral" not presented in the table.

[c]The five categories of responses were categorised into three groups in analysis: "Strongly agree/agree", "Neutral", "Strongly disagree/Disagree" in analyses. Response option "neutral" not presented in the table.

[d]Marital status, dichotomous variable. The category not married/single do not include participants who reported that they were separated/divorced or widowed.

[e]Included health facilities were primarily offering public health care.

[f]Performing ultrasound examinations.

[g]Included in analysis were only participants reporting a role in clinical decision-making.

The Venn diagram in Fig 2 illustrates agreements (disagreements and neutral responses not included in the figure) for the two statements "Fetal health interests should be better protected by law" and "Maternal health interests should always be prioritised over fetal health interests in care provided" (all health professional categories included). Venn diagrams selecting obstetricians and midwives, respectively, showed similar patterns (not presented). Fig 2 illustrates that participants did not perceive a contradiction when agreeing to both statements. Fig 3 presents the sizes and intersections between the three statements: "Fetal health interests should be better protected by law (agreement), "The fetus is never a patient, only the pregnant woman can be the patient" (disagreement), and "The delivery sometimes has to be postponed in order to improve fetal outcome, although the pregnant woman may be at risk" (agreement). The two latter statements largely coincide with the first statement. Fewer than half of the participants

**All participants**

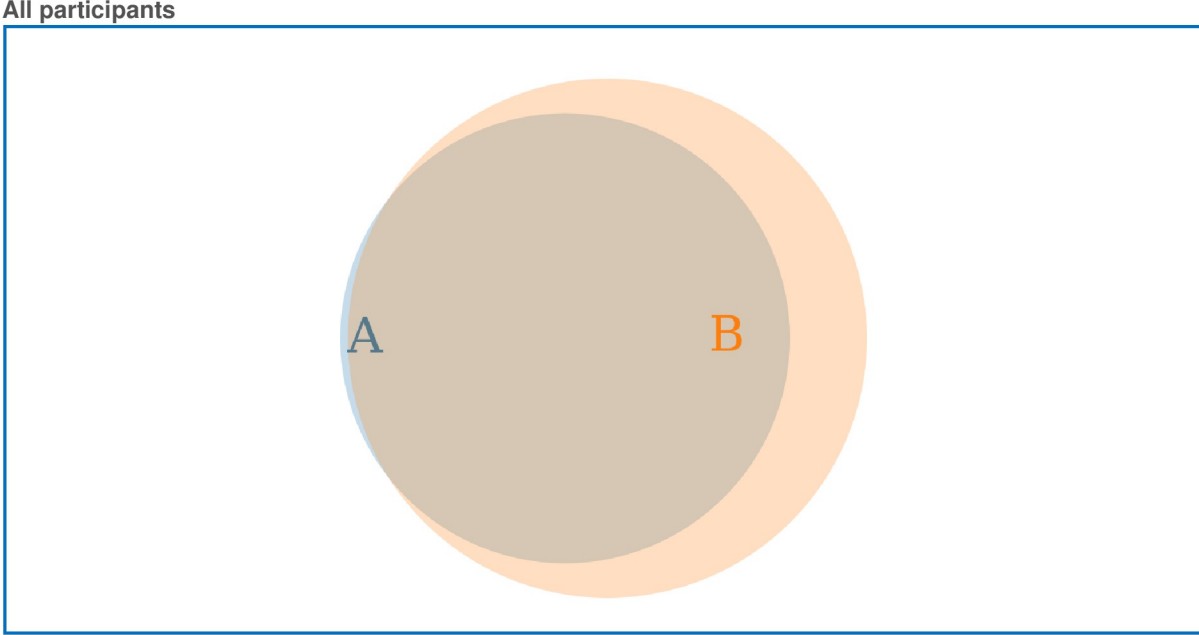

A (n=427); B (n=569)
A and B intersection is corresponding to n=423

**Fig 2. Venn diagram presenting the numbers and proportions of all that agreed/strongly agreed with the two statements.** "Maternal health interests should always be prioritised over fetal health interests in care provided; A: blue area), "Fetal health interests should be better protected by law; B: apricot area).

who disagreed with the statement "The fetus is never a patient, only the pregnant woman can be the patient" also agreed that "The delivery sometimes has to be postponed to improve fetal outcome, although the pregnant woman may be at risk".

## Discussion

Our study aimed to explore how various background factors, including the use of obstetric ultrasound, influence the experiences and perspectives of Vietnamese health professionals on maternal and fetal health interests in maternal care. Previous research, including our own, has highlighted the ethical dilemmas that arise from the special interdependent relationship between the pregnant woman and her fetus [7, 15–23, 25–28]. In this study, most participants acknowledged the fetus as a person whose personhood evolves gradually. Specifically, the majority believed that personhood begins at conception for the fetus, yet they did not agree with the notion that only the pregnant woman, and not the fetus, can be considered a patient. Additionally, most participants strongly felt that maternal health interests should always take precedence over fetal health interests in care.

The use of obstetric ultrasound undoubtedly brings many maternal and fetal health benefits. However, it has been argued that continuing advances in the medico-technical field have led to an increasing medicalisation of pregnancy and childbirth [20, 29] and that both women and healthcare workers appear to overestimate the diagnostic power of obstetric ultrasound [30, 31] while underestimating the importance of regular ANC visits [5, 7]. Visual ultrasound technology can be linked to the "personification" of the fetus [29, 32]. This "personification" has also contributed to individual rights being increasingly attributed to the fetus [29] and may diminish the central maternal role in pregnancy and maternal autonomy [33]. In contrast, a routine ultrasound examination during pregnancy has been described as 'meeting and

**All participants**

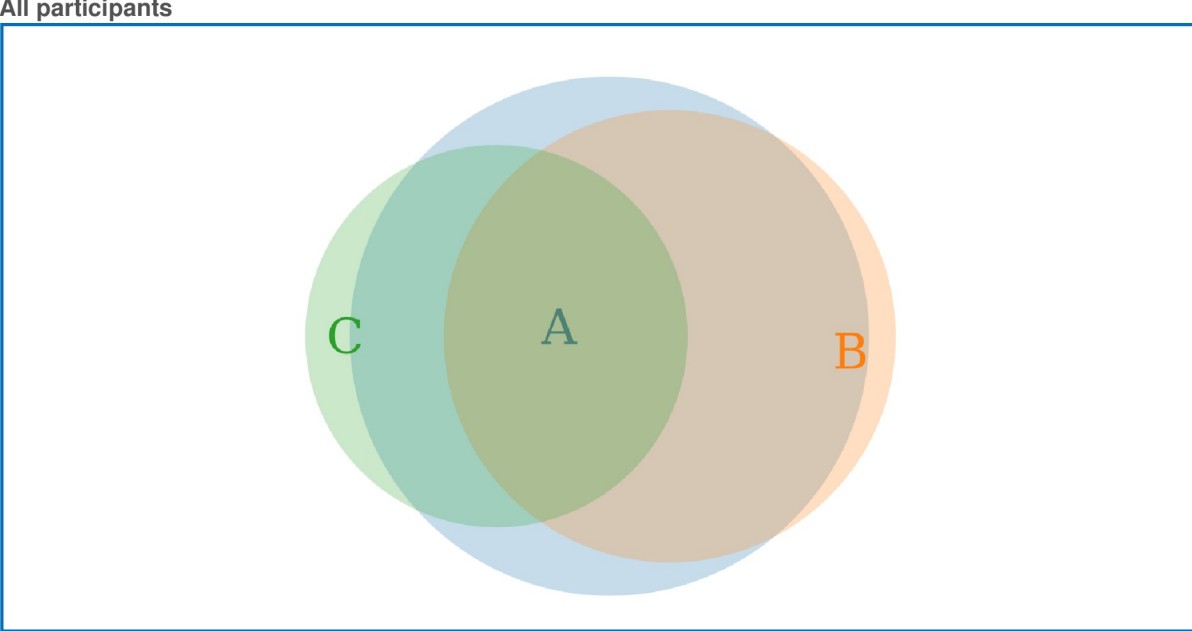

A (n=625); B (n=474); C (n=339)

A and B intersection is corresponding to n=466
A and C intersection is corresponding to n=323
B and C intersection is corresponding to n=190
A and B and C intersection is corresponding to n=181

**Fig 3. Venn diagram presenting the numbers and proportions of all participants that agreed/strongly agreed or disagreed/strongly disagreed (specified) with the following three statements.** "Fetal health interests should be better protected by law (agreement); A: blue area), "The fetus is never a patient, only the pregnant woman can be the patient (disagreement); B: apricot area), and "The delivery sometimes has to be postponed in order to improve fetal outcome, although the pregnant woman may be at risk (agreement); C: green area).

connecting with the baby' by expectant parents. The ultrasound examination may thus be considered an important step towards parenthood [34, 35].

Whether the fetus fulfils the requirements of personhood has long been debated among philosophers and others [10, 12], where the existing tension between medical ethics and the law may not always be avoidable [36]. Previous research demonstrates that practitioners may consider the fetus a person, as well as a patient [11] and that the fetus acquires its personhood gradually [37], a finding further substantiated by our study. The visualisation of the fetus in the uterus with the assistance of modern technology, combined with increased capacity for fetal treatment, has led health professionals to conceptualise the fetus as a patient, owed the same obligations as other patients [33, 38–41]. It has been proposed that intrapartum management counselling of the pregnant woman–and any individuals she chooses to involve–should adhere to two ethical principles: the fetus as a patient, deserving appropriate medical care, and the pregnant woman as the ultimate decision maker, holding primary authority over health decisions. By integrating these principles, healthcare providers can offer comprehensive and respectful counselling, aiding informed decision-making during labour and delivery [42]. It has been demonstrated, however, that healthcare providers are inconsistent in their response to pregnant women making the final decision regarding the care received [43]. Furthermore, clinicians may not always be aware of the influences of their beliefs and values in the clinical management of pregnant women [44]. As competent adults, it has been argued that women

have the right to determine what happens to their bodies, and interference without consent breaks the ethical principle of autonomy [45].

International guidelines for obstetricians and midwives emphasise respecting the autonomy of pregnant women [46, 47], advocating for a legal framework that prioritises the woman's decision-making rights [36]. Despite these guidelines, inconsistencies exist in the responses of physicians and midwives towards pregnant women as the final decision-makers in the care received [44]. These discrepancies highlight differing attitudes and beliefs among healthcare providers regarding pregnant women's rights and their own legal accountability in the clinical management of pregnant women [43, 44]. While most pregnant women strive to improve their chances of having a healthy baby, conflicts can arise when maternal health interests do not align with fetal health interests [38, 48]. In our study, there was broad agreement among participants that maternal health interests should always be prioritised over fetal health interests in the care provided. Additionally, there was a consensus that fetal health interests are given increasing weight in decision-making as gestation progresses. Most participants also supported the notion that fetal health interests should be better protected by law. Interestingly, participants did not perceive a contradiction in agreeing that fetal health interests should be better protected by law while also maintaining that maternal health interests should always take precedence in the care provided. This finding suggests a nuanced understanding among health professionals of balancing maternal and fetal rights without compromising maternal autonomy. Nevertheless, assigning the fetus greater status might imply decreased maternal autonomy, which would be an unwarranted development [49]. Therefore, maternal reproductive rights and human rights need close monitoring and potentially further safeguarding in the future.

Congenital fetal malformations are often seen as a burden to families and society in Vietnam [6], prompting Vietnamese authorities to recommend premarital health check-ups, especially for those exposed to toxic chemicals like Agent Orange [50, 51]. The legalisation of abortion in Vietnam in 1959 and the subsequent rise in abortion incidence following the introduction of the one-to-two-child policy in 1988 [52], reflect the country's evolving reproductive health policies. Currently, Vietnamese law permits legal abortion up to 22 weeks of gestation [6], with no upper limit for abortions due to fetal malformations [46]. These legal frameworks underscore the importance of addressing ethical considerations in reproductive health, ensuring that maternal autonomy is respected while also considering the evolving status of the fetus within the medical and legal contexts. Deeper ethical discussions of the status of the fetus as a person, its personhood, and its designation as a patient would, therefore, be valuable as these concepts are critical in shaping clinical practices and policies related to prenatal care and reproductive rights. Defining the fetus's personhood and patient status has profound implications for maternal autonomy, legal accountability and the ethical obligations of healthcare providers. A comprehensive ethical analysis could contribute to a more nuanced balance between respecting the rights and health interests of the pregnant woman and the medical and moral considerations surrounding the fetus–and thus ensure that policies and practices are aligned with both ethical principles and the evolving societal values concerning reproductive health.

## Methodological considerations

A strength of this study was that health professionals from different levels of the health care system in the Hanoi region participated, representing urban, semi-urban and rural areas. In addition, the research team included two Vietnamese researchers familiar with the setting and the healthcare system. Different research areas, such as obstetrics, venereology, nursing, midwifery, public health, and statistics, were represented among the authors, contributing to

various perspectives. Limitations of the study have previously been presented in detail [14]. In brief, there was a risk of losing the intended meaning of questions and statements while translating the questionnaire from English to Vietnamese. However, a back-translation of the questionnaire was performed to reduce this risk, resulting in minor adjustments of some words. Despite aiming for an even distribution among health professionals, in the end, more midwives were included compared to physicians.

## Conclusions

Our results indicate that a large proportion of health professionals in Vietnam assign the fetus the status of being a person, where personhood is seen as gradually evolving during pregnancy. To a large extent, the fetus is considered a patient with its own health interests, although a majority give priority to maternal health interests. Health professionals appear to favour increased legal protection of fetal status. Strengthening the legal position of the fetus might have adverse implications for maternal autonomy and would need to be monitored carefully were this to occur. Maternal reproductive rights might need to be further safeguarded.

## Acknowledgments

We thank the participating health professionals for sharing their time and experiences and for the support of the heads of the Departments of Obstetrics and Gynecology at the selected health facilities. Joseph Ntaganira passed away before the submission of the final version of this manuscript. Cecilia Bergström accepts responsibility for the integrity and validity of the data collected and analysed. We express our sincere gratitude to the late for his invaluable contributions. His involvement in the design, conceptualisation, and data collection was instrumental in shaping the foundation of this research project. Thanks go to the data collectors Do Nam Khanh, Nguyen Huyen Tram, Le Vu Thuy Huong and Nguyen Thi Hue for their excellent performance in the data collection. We acknowledge the support received from Hanoi Medical University (Hanoi, Vietnam) and Umeå University (Umeå, Sweden).

## Author Contributions

**Conceptualization:** Ingrid Mogren, Pham Thi Lan, Ho Dang Phuc, Sophia Holmlund, Rhonda Small, Joseph Ntaganira, Jean Paul Semasaka Sengoma, Hussein Lesio Kidanto, Matilda Ngarina, Cecilia Bergström.

**Data curation:** Ingrid Mogren, Pham Thi Lan, Ho Dang Phuc, Sophia Holmlund, Hussein Lesio Kidanto, Cecilia Bergström.

**Formal analysis:** Ingrid Mogren, Cecilia Bergström.

**Funding acquisition:** Ingrid Mogren.

**Investigation:** Ingrid Mogren, Pham Thi Lan, Ho Dang Phuc, Sophia Holmlund.

**Methodology:** Ingrid Mogren, Pham Thi Lan, Ho Dang Phuc, Sophia Holmlund.

**Project administration:** Ingrid Mogren, Pham Thi Lan, Sophia Holmlund, Cecilia Bergström.

**Resources:** Ingrid Mogren, Pham Thi Lan, Ho Dang Phuc.

**Supervision:** Ingrid Mogren, Pham Thi Lan, Ho Dang Phuc, Sophia Holmlund, Cecilia Bergström.

**Validation:** Ingrid Mogren, Pham Thi Lan, Ho Dang Phuc, Cecilia Bergström.

**Visualization:** Ingrid Mogren, Cecilia Bergström.

**Writing – original draft:** Ingrid Mogren, Cecilia Bergström.

**Writing – review & editing:** Ingrid Mogren, Pham Thi Lan, Ho Dang Phuc, Sophia Holmlund, Rhonda Small, Joseph Ntaganira, Jean Paul Semasaka Sengoma, Hussein Lesio Kidanto, Matilda Ngarina, Cecilia Bergström.

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
