## [Decision Letter · Decision Letter 0]

29 May 2024

PONE-D-23-37905Vietnamese health professionals’ views on the status of the fetus and maternal and fetal health interests: A regional, cross-sectional study from the Hanoi areaPLOS ONE

Dear Dr. Bergström,

Thank you for submitting your manuscript to PLOS ONE. After careful consideration, we feel that it has merit but does not fully meet PLOS ONE’s publication criteria as it currently stands. Therefore, we invite you to submit a revised version of the manuscript that addresses the points raised during the review process.

We look forward to receiving your revised manuscript.

Kind regards,

Patrick Ifeanyi Okonta, MBBCh, MPH, FWACS, FMCOG, MD, DRH

Academic Editor

PLOS ONE

Reviewers' comments:

Reviewer's Responses to Questions

**Comments to the Author**

1. Is the manuscript technically sound, and do the data support the conclusions?

Reviewer #1: Partly

Reviewer #2: Yes

2. Has the statistical analysis been performed appropriately and rigorously? 

Reviewer #1: Yes

Reviewer #2: Yes

3. Have the authors made all data underlying the findings in their manuscript fully available?

Reviewer #1: Yes

Reviewer #2: Yes

4. Is the manuscript presented in an intelligible fashion and written in standard English?

Reviewer #1: No

Reviewer #2: Yes

5. Review Comments to the Author

Reviewer #1: 1. Some grammatical errors noted. Go through the manuscript once more and may need to engage a language editor to make some of the sentences easily comprehensible by the reader.

2. The tables were not presented in a reader-friendly manner making it very difficult to appreciate. I suggest contents of the tables are reduced to bring out only more relevant data in a concise way.

3. Responses to the questions should have been mutually exclusive, where a YES answer to one negates the others. In its current form, it becomes very difficult to make distinctions e.g when does the fetus become a person - the question cannot be answered when responders select multiple answers to the same question. This should be corrected.

4. The discussion leaves much more to be desired. I believe there should be more of ethical discourse involved in defining the fetus's person, personhood and patient's status. Also a brief review of Vietnamese laws as they pertain to the subject of interest might also have been apt.

Reviewer #2: Thank you for your interesting and well-researched original article. I have a few corrections to help improve clarity

a. Page 3 line 35: should read purposively

b. Page 6 line 104: ‘based on in Vietnam’—please clarify

c. Page 29 lines 486-488: Please clarify--do you mean "her relatives after obtaining her consent"

6. PLOS authors have the option to publish the peer review history of their article (what does this mean?). If published, this will include your full peer review and any attached files.

Reviewer #1: No

Reviewer #2: **Yes: **Ijeoma Victoria

---

## [Author Response · Author response to Decision Letter 0]

14 Aug 2024

We appreciate the reviewers’ comments addressing limitations in the submitted manuscript. The following letter includes comments on the reviewers’ reports on the submitted manuscript to PLOS ONE.

Reviewer #1: 

1. Some grammatical errors noted. Go through the manuscript once more and may need to engage a language editor to make some of the sentences easily comprehensible by the reader.

In our efforts to enhance readability, the manuscript underwent thorough language revision, aiming to improve its reader-friendliness.

2. The tables were not presented in a reader-friendly manner making it very difficult to appreciate. I suggest contents of the tables are reduced to bring out only more relevant data in a concise way.

Thank you for your feedback on the presentation of our tables. We understand the importance of clear and accessible information. In response to your suggestions, we have revised the tables to enhance their readability. This involved presenting data in a more concise format. We believe these changes will make the information easier to comprehend and appreciate. We value your input and hope you find the updated tables more user-friendly.

3. Responses to the questions should have been mutually exclusive, where a YES answer to one negates the others. In its current form, it becomes very difficult to make distinctions e.g when does the fetus become a person - the question cannot be answered when responders select multiple answers to the same question. This should be corrected.

Thank you for your comment. We appreciate your insights on the complexity surrounding the status of the embryo/fetus. In the section “Statements on views of the fetus” (Table 1), the views were analysed in ascending chronological order, assuming that if the fetus were considered a person at an earlier chronologic stage, then personhood was presumed to be continuous during the remaining course of the pregnancy and at birth. By recognising the diverse stages and situations where individuals may view the embryo/fetus as a person, we developed various statements to reflect the spectrum of beliefs, as detailed in Table 1 and shown for convenience below: 

• The fetus is a person from the time of conception

• The fetus is a person from the time heartbeats are detected

• The fetus is a person from the time the pregnant woman experiences fetal movements

• The fetus is a person when it can survive outside the uterus

• The fetus is a person when the pregnant woman considers it to be a person

• The fetus is a not a person until it is born

For clarity and consistency, we chose to use the term “fetus” throughout the questionnaire. Our aim was to capture the range of perspectives regarding the fetus as a person, acknowledging that these viewpoints are not mutually exclusive. These statements were included in the questionnaire that was piloted across six countries with health professionals who were also eligible participants in the qualitative part of the CROCUS Study. Pilot participants completed these items with no issues raised about their relevance or appropriateness. Therefore, they were included in the final questionnaire and were not reported as problematic by participants in the study.

4. The discussion leaves much more to be desired. I believe there should be more of ethical discourse involved in defining the fetus's person, personhood and patient's status. Also a brief review of Vietnamese laws as they pertain to the subject of interest might also have been apt.

In response to the reviewer's feedback, we have revised the discussion section to incorporate a more comprehensive ethical discourse on the fetus's status as a person, personhood and patient. These revisions aim to provide a deeper exploration of these ethical concepts and their relevance to our study. Furthermore, we have included a brief review of Vietnamese laws regarding the fetus's legal status and rights. This contextual addition ensures that our discussion aligns with the ethical and legal frameworks relevant to our subject of interest. These revisions are intended to enhance the discussion section's overall coherence and relevance, thereby facilitating a better understanding of the study's implications in ethical and legal contexts.

Reviewer #2: Thank you for your interesting and well-researched original article. I have a few corrections to help improve clarity

a. Page 3 line 35: should read purposively

This has been corrected. 

b. Page 6 line 104: ‘based on in Vietnam’—please clarify

This must have been a typo. It now reads: …when providing maternity care in Vietnam.

c. Page 29 lines 486-488: Please clarify--do you mean "her relatives after obtaining her consent"

We agree that this sentence needs further clarification. For convenience, please see below.

It has been proposed that intrapartum management counselling of the pregnant woman – and any individuals she chooses to involve – should adhere to two ethical principles: the fetus as a patient, deserving appropriate medical care, and the pregnant woman as the ultimate decision maker, holding primary authority over health decisions. By integrating these principles, healthcare providers can offer comprehensive and respectful counselling, aiding informed decision-making during labour and delivery.

---

## [Editor Report · Decision Letter 1]

23 Aug 2024

Vietnamese health professionals’ views on the status of the fetus and maternal and fetal health interests: A regional, cross-sectional study from the Hanoi area

PONE-D-23-37905R1

Dear Dr. Bergström,

We’re pleased to inform you that your manuscript has been judged scientifically suitable for publication and will be formally accepted for publication once it meets all outstanding technical requirements.

Kind regards,

Patrick Ifeanyi Okonta, MBBCh, MPH, FWACS, FMCOG, MD, DRH

Academic Editor

PLOS ONE
---

## [Editor Report · Acceptance letter]

2 Sep 2024

PONE-D-23-37905R1 

PLOS ONE

Dear Dr. Bergström, 

I'm pleased to inform you that your manuscript has been deemed suitable for publication in PLOS ONE. Congratulations! Your manuscript is now being handed over to our production team.

Kind regards, 

on behalf of

Professor Patrick Ifeanyi Okonta 

Academic Editor

PLOS ONE